# Segmentation and Grade Prediction of Colon Cancer Digital Pathology Images Across Multiple Institutions

**DOI:** 10.3390/cancers11111700

**Published:** 2019-11-01

**Authors:** Saima Rathore, Muhammad Aksam Iftikhar, Ahmad Chaddad, Tamim Niazi, Thomas Karasic, Michel Bilello

**Affiliations:** 1Center for Biomedical Image Computing and Analytics, University of Pennsylvania, Philadelphia, PA 19104, USA; michel.bilello@uphs.upenn.edu; 2Department of Radiology, Perelman School of Medicine, University of Pennsylvania, Philadelphia, PA 19104, USA; 3Department of Computer Science, COMSATS University Islamabad, Lahore Campus, Lahore 54000, Pakistan; aksam.iftikhar@gmail.com; 4Division of Radiation Oncology, Department of Oncology, McGill University, Montreal, QC H3S 1Y9, Canada; ahmad.chaddad@mail.mcgill.ca (A.C.); tniazi@jgh.mcgill.ca (T.N.); 5Department of Medicine, Division of Hematology/Oncology, University of Pennsylvania, Philadelphia, PA 19104, USA; Thomas.Karasic@pennmedicine.upenn.edu

**Keywords:** hierarchical classification, gland segmentation, colon cancer detection, colon cancer grading

## Abstract

Distinguishing benign from malignant disease is a primary challenge for colon histopathologists. Current clinical methods rely on qualitative visual analysis of features such as glandular architecture and size that exist on a continuum from benign to malignant. Consequently, discordance between histopathologists is common. To provide more reliable analysis of colon specimens, we propose an end-to-end computational pathology pipeline that encompasses gland segmentation, cancer detection, and then further breaking down the malignant samples into different cancer grades. We propose a multi-step gland segmentation method, which models tissue components as ellipsoids. For cancer detection/grading, we encode cellular morphology, spatial architectural patterns of glands, and texture by extracting multi-scale features: (i) Gland-based: extracted from individual glands, (ii) local-patch-based: computed from randomly-selected image patches, and (iii) image-based: extracted from images, and employ a hierarchical ensemble-classification method. Using two datasets (Rawalpindi Medical College (RMC), *n* = 174 and gland segmentation (GlaS), *n* = 165) with three cancer grades, our method reliably delineated gland regions (RMC = 87.5%, GlaS = 88.4%), detected the presence of malignancy (RMC = 97.6%, GlaS = 98.3%), and predicted tumor grade (RMC = 98.6%, GlaS = 98.6%). Training the model using one dataset and testing it on the other showed strong concordance in cancer detection (Train RMC – Test GlaS = 94.5%, Train GlaS – Test RMC = 93.7%) and grading (Train RMC – Test GlaS = 95%, Train GlaS – Test RMC = 95%) suggesting that the model will be applicable across institutions. With further prospective validation, the techniques demonstrated here may provide a reproducible and easily accessible method to standardize analysis of colon cancer specimens.

## 1. Introduction

Distinguishing invasive cancer from premalignant dysplasia or benign proliferation is a fundamental task of pathologists examining colon specimens, and proper diagnosis and grading is essential to guide treatment. One of the primary challenges in colon histopathology is inter- and intra-observer variability that can significantly alter treatment decisions [1,2]. Review by expert pathologists with subspecialty training in gastrointestinal malignancy is the gold standard for diagnosis, but such second opinions are labor intensive, slow, and frequently unavailable in many resource-poor areas. With digitized images of histology slides becoming increasingly ubiquitous, automated digital pathology offers an expedited and viable solution to this problem [3,4,5].

Digital analysis of histology images enables extraction of quantitative morphological features, which can be used for computer-assisted grading of cancer, making the grading process more objective and reproducible [6,7]. For colorectal cancers, computer-aided diagnostic systems must analyze several morphological features of intestinal glands, including cellular architecture, gland formation, and stromal components (Figure 1). Normal colon biopsy samples have a typical and uniform glandular arrangement, but malignant tumor samples have a wide and heterogenous spectrum of disruption of standard histologic features. Our research aims to exploit such variations to address three main components of automated colon cancer diagnosis: (i) Automated gland segmentation, (ii) colon cancer detection, and (iii) colon cancer grading. It is important to note that cancer detection, in this study, refers to classifying between normal and malignant samples, whereas cancer grading, refers to further breaking down the malignant samples into different cancer grades.

Most of the published research on gland segmentation demarcates glandular boundaries, and disregards the segmentation of internal glandular components. Moreover, the existing studies on automated analysis of histologic colon cancer specimens are based on global image features, and such analyses may miss important local details that may alter the diagnosis or treatment recommendations [3,8,9,10,11]. Prior automated systems have also typically focused on separate analyses for segmentation, cancer detection, and cancer grading [12,13,14], and have rarely developed end-to-end pipelines.

In this paper, we propose a robust gland segmentation method for segmentation of glands and their internal structures such as epithelial cells, nuclei and lumen. We also propose a novel hierarchical classification method exploiting features at multiple image scales for colon cancer detection and grade prediction. We aim to build an end-to-end computational pathology pipeline (Figure 2) for histologic colon cancer detection and grade prediction by incorporating gland segmentation, cancer detection, and grading into a single automated analysis. This study contributes in five different ways: A robust gland segmentation method that exploits the organizational appearance of colon glands to demarcate gland boundaries and to segment internal glandular structures, including epithelial cells, nuclei and lumen, and yields promising results for multiple datasets. The radiomic features extracted from the segmented structures were later used in the classification.A multi-scale feature extraction mechanism comprising: (i) Gland-based features: extracted from individual glands, which are segmented by the proposed gland segmentation method, (ii) local-patch-based features: computed from randomly-selected patches of each histologic colon image, and (iii) image-based features: calculated by considering whole image as one unit and capturing global details.Novel gland-based features, which encode morphometric properties of epithelial cells and lumen and combine them with the spatial distribution of the nuclei in relationship to stroma and lumen.A multi-level hierarchical classification methodology to divide large-scale classification problem into a set of small-scale and easier-to-solve problems. In the proposed framework, the first-level classification produces probability estimates based on each feature type (image-, gland-, and local-patch-based features), and the second-level is an ensemble of various support vector machine (SVM) classifiers trained on the probability estimates generated by the first-level.Demonstration of the generalizability of the proposed pipeline across multiple datasets.

### 1.1. Related Work

This section presents a review of contemporary literature in the three research directions on colon cancer diagnosis, which have also been investigated in the current work, i.e., automated gland segmentation, colon cancer detection and grading.

Glandular structures in histopathology images can be segmented either by using generic [15] or specialized methods [16,17,18]. Graph-based methods, which rely on generating graphs from glandular structures, have been the most common method for gland segmentation. For example, Demir et al. decomposed the glandular structures in their primitive objects and utilized the organizational properties of these objects instead of traditional pixel-based properties [16]. The approach relies on a region growing procedure, where the gland seeds are determined based on a graph constructed from the nucleus and lumen objects. The seeds are grown using another object-graph constructed on the nucleus objects alone. The final boundary of glands is obtained based on the locations of the nucleus objects and a false-positive elimination process based on information of the segmented grown regions. Similarly, Fu et al. proposed a graph-based GlandVision algorithm [19]. Using the random field modelling in the polar space, the gland contours were extracted based on an inference strategy that approximates a circular graph using two chain graphs. Then, a support vector regressor based on visual features was used to verify that the extracted contour belonged to a true gland.

Some recent research efforts have also employed deep neural networks for gland segmentation. To this end, Kainz et al. [18] presented a deep convolutional neural network based pixel classifier for semantic segmentation of colon gland images. Two 7-layer convolutional neural networks (CNNs) were used to predict whether individual pixels belonged to normal or malignant glands. These predictions were then normalized based on weighted total variation using a figure-ground segmentation approach. Wenqi et al. [20] also employed fine-tuned CNNs for segmenting glandular structures, but combined them with an SVM classifier based on traditional radiomic features. Recently, for gland segmentation, Chen et al. proposed a deep contour-aware network that uses a unified multi-task learning framework, exploits multi-level contextual features, and employs an auxiliary supervision method to solve the problem of vanishing gradients [21]. Graham et al. proposed a CNN that counters the information loss incurred in max-pooling layers by re-introducing the original image at multiple points within the network [17]. They used atrous spatial pyramid pooling with varying dilation rates for preserving the resolution and multi-level aggregation, and introduced random transformations during test time for an enhanced segmentation result that concurrently generates an uncertainty map and highlights ambiguous areas. We aim to improve upon this prior art by proposing a gland segmentation algorithm that not only delineates gland boundaries, but also demarcates the internal glandular structures using a multi-step process based on the geometrical/morphological properties of the structures.

Several radiomic methods exist in the literature to distinguish benign and malignant colon lesions. For example, Masood et al. [22] investigated local-binary-patterns along with Gaussian SVM to produce reasonable classification results. Classification accuracy can be improved using an ensemble of different classifiers and hybrid combinations of discriminative features. For example, Rathore et al. [3] employed different ensemble classifiers such as rotation boost, rotation forest and random forest on a hybrid feature set, comprising white run-length and area features, and achieved better classification accuracy than from any individual classifier or feature. Similarly, combining a textural analysis of color components with a histogram-of-oriented-gradients improved classification performance [23]. Also, chaddad et al. proposed several radiomic pipeline to evaluate the continuum of colorectal cancer using various types of shape and texture features with multi classifier models [12,13,14]. Considered the CNN models, the classification accuracy was improved compared to the conventional classifier models [13].

Similar to automated gland segmentation, some researchers have employed graph-based techniques to standardize colon cancer grading. Altunbay et al. [9] employed the characteristic of circularity in shape of pink, purple, and white clusters of colon biopsy images. They applied a circle finding algorithm on these clusters and computed discriminative features on a graph generated from circular objects in these clusters. They validated this technique by detecting colon cancer from colon biopsy images and discriminating different cancer grades with high accuracy. Ozdemir et al. [10] presented a similar technique based on graph creation from the three clusters of colon biopsy images of normal subjects. These test graphs were compared with training graphs to determine whether tissues were normal or malignant based on the extent of correlation with the test graph. In contrast, Rathore et al. [11] used lumen circularity, convexity, concavity, and ratio of lumen area to the size of image and white cluster as features, which improved the accuracy of cancer grading. In a recent study, Awan et al. measured the shape of glands with a novel metric that they called the “best alignment metric” (BAM). A SVM classifier was then trained using shape features derived from BAM that yielded an accuracy of 91% in a three-class classification into normal, low grade cancer, and high grade cancer [24]. A comprehensive review further describes colon cancer segmentation, detection and grading techniques [7].

Despite the significant advances in the past two decades, end-to-end computational pathology pipelines have rarely been developed by the researchers. Our paper aims to bridge this gap and provides an end-to-end computational pathology pipeline for histologic colon cancer detection and grade prediction by incorporating gland segmentation, cancer detection, and grading into a single automated analysis.

## 2. Results

### 2.1. Performance of The Proposed Gland Segmentation Method

The segmentation results of the proposed algorithm are visually illustrated for various normal and malignant samples (Figure 3). These results demonstrate that the proposed algorithm successfully identifies glandular boundaries and properly segments glands across irregularly arranged malignant samples. The corresponding quantitative results are shown in Table 1 for two different datasets, gland segmentation (GlaS) challenge dataset, and Rawalpindi Medical College (RMC) dataset.

### 2.2. Performance of The Proposed Cancer Detection and Grading Method

Classification rates obtained using the individual features are summarized in terms of classification accuracy, sensitivity, specificity, and Matthew’s correlation coefficient (MCC) for cancer detection and respective grading (Table 2). Our predictive model’s accuracies in correctly classifying the normal and malignant images on the GlaS and RMC datasets were 93.7% and 92.1% for gland-based features, 93.1% and 91.5% for patch-based features, and 92.5% and 90.9% for image-based features, respectively. Similarly, the cross-validated 3-class accuracy of cancer grading on the GlaS and RMC datasets was 90.5% and 92.5% for gland-based features, 89.5% and 91.5% for patch-based features, and 89.7% and 90.7% for image-based features, respectively. No statistically significant differences of the feature extraction methods between the two datasets by McNemar Test were noted.

The various constituents of the meta-classifier were trained based on the probability estimates generated by the first-level classification, and their output was combined using majority voting to get final predictions (Table 3). The cross-validated 2-class accuracy of cancer detection was 98.3% for the GlaS dataset and 97.6% for the RMC dataset. Similarly, the cross-validated 3-class accuracy of cancer grading was 98.6% both for GlaS and RMC datasets. All the individual classifiers performed reasonably well for both cancer detection and grading tasks, however, an optimal combination of many complementary imaging features/classifiers through a meta-classifier achieved even higher accuracy, which is statistically significant (*p* < 0.01) at 95% confidence interval. Furthermore, comparing the assessment of individual features/classifiers with the multivariate results of combined features and classifiers reveals that subtle individual features can be synthesized in an index of higher distinctive performance.

A receiver operating characteristic (ROC) analysis was performed to verify the reliability of classification results. Area under the curve (AUC) was 0.95 and 0.99 for cancer detection of the RMC and GlaS datasets, respectively, and 0.98 and 0.96 for cancer grading of the RMC and GlaS datasets, respectively (Figure 4).

### 2.3. Performance of the Proposed Cancer Detection and Grading Method across Different Institutions

To determine the applicability of our radiomic analysis across datasets, we trained the two-stage classification framework on one dataset and tested it on the other, and vice versa (Table 4). For cancer detection, a model trained on the RMC dataset and tested on the GlaS dataset yielded an accuracy of 94.5%, while a model trained on GlaS and tested on RMC yielded an accuracy of 93.7%. For cancer grading, an accuracy of 95.0% was observed for both training and testing pairs.

### 2.4. Interpretation of Features of Normal/Malignant Colon Tissues

To obtain a deeper understanding of the information (features) used by SVM to provide predictions, boxplots of some predictive features were generated by dividing the patient cohort into 2 groups according to the normal/malignant status. We have selected top-most 5 features based on the weight assigned by the classifier, and compared the values of these features in normal and malignant groups. Boxplots of these features based on data from all the patients were created with the group type on x-axis and its value on y-axis (Figure 5). The analysis revealed that each tissue type had a unique set of radiographically interpretable features associated with it. The main findings from comparing the features of different tissue types are as follows:Lower entropy values in normal colon tissue images (*p =* 5.6 × 10^−5^), suggestive of more uniformity and proper organizational structure of tissue compared to malignant colon tissue that shows very high entropy, indicative of more randomness and lack of any organizational structure;Lower contrast values in normal colon tissue (*p =* 2.1 × 10^−6^), which again point towards coherent and properly organized tissue. On the other hand, the higher contrast values in malignant colon tissues show the lack of proper organizational structure;Lower values of standard deviation of distances of cell nuclei from the centroid of lumen for normal colon tissue (*p =* 8.6 *×* 10^−4^) are suggestive of the fact that cell nuclei are almost equally spaced from the centroid of the lumen, whereas the same is not true for malignant colon tissue where cell nuclei are at random distances from lumen;Higher values of eccentricity (*p =* 2.1 × 10^−3^) and compactness (*p =* 5.8 × 10^−4^) in normal colon tissues, consistent with the well-defined shape of normal colon tissue. On the contrary, the malignant colon tissues exhibit lower compactness and eccentricity, indicative of the migratory and deeply infiltrated tissue.

None of the individual features are sufficient to distinguish normal from malignant tissue but multivariate analysis through machine learning accurately classifies benign and malignant samples.

## 3. Discussion

In this paper, we have demonstrated accurate identification and grading of malignant colon cancer samples using an automated computational pathology pipeline.

The segmentation method proposed herein provided an accurate demarcation of glandular regions of the colon tissue. We want to mention that despite the prior development of several deep-learning based gland segmentation algorithms, we proposed an additional algorithm because the proposed algorithm not only demarcates boundaries but also divides glandular region into its constituent elements, that we later quantify in terms of features to use in the cancer detection and grading steps. Also, in the absence of ground-truth of internal glandular regions, training deep learning algorithms was not possible, therefore, we adopted an alternative approach comprising a series of image processing steps to segment internal glandular regions. Moreover, there are several performance advantages of the proposed algorithm over deep learning approaches. First, the processing time of this algorithm is lower than that of computationally expensive deep learning algorithm. Second, this algorithm does not need any training, and can be applied on any incoming specimen without having a pre-trained model in place. Third, the method does not need any sophisticated hardware to run, and can run on desktop machines having only CPU. The performance of the proposed algorithm on GlaS dataset (F-Score = 0.89, Dice = 0.87) for the task of outer glandular boundary segmentation is also in par with the existing methods. For example, the best performing method in MICCAI 2015 gland segmentation contest [21] reported F-Score of 0.887, and another recent study [17] reported F-Score of 0.844 and Dice of 0.836 on Test A, and F-Score of 0.914 and Dice of 0.913 on Test B datasets, respectively, released as part of MICCAI 2015 gland segmentation contest.

Our multi-feature hierarchical classification method consistently and substantially improved upon the classification performance of the individual-features-based classification methods. Similarly, the proposed ensemble frameworks, either the ensemble for patch-based features at the first-level of classification or the ensemble for meta-classifier at the second-level of classification, reinforced the classification accuracy of individual classifiers and yielded better prediction estimates. Overall, our methods achieved a high accuracy for cancer detection (RMC = 97.6%, GlaS = 98.3%) and grading (RMC = 98.6%, GlaS = 98.6%). Importantly, the features highlighted as most discriminative by the machine learning algorithm including entropy, cellular eccentricity, and nuclear displacement, are the same visual features relied upon by histopathologists to distinguish between normal and malignant tissues. Our findings are consistent with the previous literatures. For example, Chaddad et al. showed that the feature of entropy is able to discriminate between four type of pathology tissues [12]. It was also observed in another study [25] that the measure of eccentricity was a good predictor of intraepithelial neoplasia, which is a precursor state for full-blown carcinoma.

### 3.1. Validation of the Proposed Method across Different Datasets

To confirm that our methods would be applicable across multiple institutions, we trained our model on a dataset from one institution and tested it on a dataset from another institution. Previous studies, either on demarcation of malignant and benign tissue regions, or on cancer detection and grading have not undergone such validation [11,16,19,22,26,27]. Our findings (Table 3 and Table 4) support the notion that the combination of features and machine learning classification proposed in this study allows robust classification of colon cancer datasets arising from multiple institutions, even if a new dataset comes from an institution that was not part of the training sample. The validation of our methods across datasets strongly suggests that our model will perform well in routine clinical settings where samples are much more diverse than in controlled experimental settings.

### 3.2. Importance of the Study

The computational pathology pipeline described in this paper addresses many of the current barriers in clinical histopathology. This method offers a standardized approach to resolve intra- and inter-observer variability amongst pathologists and is easily implemented even in low resource settings with available digital scanning tools. Importantly, although this study is focused on colon cancer, the same approach could also be used for other types of cancer. For example, automated analysis may improve grading in neuroendocrine tumors, where treatment decisions are heavily reliant on accurate grading. Similarly, the speed and reproducibility of this method may improve reliability of grading in heterogeneous samples. One additional application may be to identify tumor origin by facilitating comparisons of multiple biopsy specimens from the same patient to determine the morphological concordance between a metastatic site and a primary tumor.

### 3.3. Limitations and Future Work

Our study has several limitations. First, the proposed segmentation method has specially been designed to segment the glandular regions, and is not suitable for regional separation of benign and malignant tissue. However, the method could be tailored, with appropriate modifications, to be used for regional demarcation of benign and malignant regions. Second, since our method is specially designed to decode the morphology of glands, it cannot accurately quantify the features of extremely deformed glands in very poorly-differentiated specimens. Another limitation of our study is that we used retrospective data; a prospective dataset comparing our methods to standard histopathological review would lend further validity to our model.

Beyond the scope of this work, there are several important future research directions. Previous machine learning studies have shown strong association between the CT imaging characteristics and survival of colorectal cancer patients [28] and molecular subtypes [29]. We aim to examine whether the analysis of histologic features demonstrated here can similarly predict clinical outcomes and genomic aberrations. A systematic analysis of these characteristics with the help of gland segmentation as part of automatic image analysis framework could lead to a better understanding of the relevant cancer biology as well as bring precision and accuracy into assessment and prediction of the outcome of the cancer.

## 4. Materials and Methods

### 4.1. Datasets

The pathology data were obtained from two independent sources: (i) 174 images, acquired at 10× magnification factor, were collected from pathology department of RMC, Rawalpindi, Pakistan. These images were subdivisions of 68 hematoxylin and eosin stained slides of colon resection specimens, with each slide belonging to a different patient. The hematoxylin stains cell nuclei purple, and eosin stains the extracellular matrix and cytoplasm pink, with other structures such as lumen and epithelial cells taking on shades very close to white color. The images were labeled as normal (*n* = 82) or malignant (*n* = 92) by an expert pathologist, and malignant images were further categorized into three different grades (well-differentiated = 44, moderately-differentiated = 25, and poorly-differentiated = 23) [27]; (ii) 165 images, acquired at 20× magnification factor, were collected from the GlaS contest [30]. These images were captured from 16 hematoxylin and eosin stained slides of colon resection specimens, each from a different patient. Images were labelled by an expert pathologist as normal (*n* = 74) or malignant (*n* = 91), and malignant images were further distributed into three different grades (moderate = 47, moderate-to-poor = 20, poor = 24) [31]. To allow for comparison between the two data sets, we converted these three categories into well-, moderately-, and poorly-differentiated grades. Contrast enhancement was applied to decrease imaging artifacts. Following contrast-enhancement, images were converted to gray-scale for further analysis.

### 4.2. The Gland Segmentation Algorithm

A multi-step image processing method (Figure 6) was used to delineate colon glands in biopsy images:

#### 4.2.1. Clustering of Tissue Components

Each image was divided into white, pink and purple clusters using the K-means algorithm. A morphological operation of erosion was performed on the clusters. 

#### 4.2.2. Ellipse Fitting

An ellipse-fitting algorithm [27] was used to identify purple-colored nuclei and white-colored epithelial cell cytoplasm and glandular lumina.

#### 4.2.3. Discrimination of True and False Glandular Components

To improve the ellipse-fitting algorithm, hierarchical clustering was applied to discriminate between true glands and false positives. Four co-centric ellipses of different orientations were overlaid on each detected object, and the following features were analyzed: (i) Total number of detected objects; (ii) average Euclidean distance between the centers of objects; (iii) mean and standard deviation of areas and compactness of objects; and (iv) ratio of purple area to white/pink area (for purple objects), and ratio of white area to pink/purple cluster (for white objects). Hierarchical clustering, when applied separately on the features of white and purple objects, divided the purple objects into two classes: objects on the boundary of the glands (true cell nuclei) and false positive objects scattered within stroma. Similarly, the clustering of the white objects divided the objects into true epithelial cell cytoplasm and glandular lumina, and false positive white areas outside the gland. 

#### 4.2.4. Lumen Detection

Following the identification of epithelial cell cytoplasm and gland lumina in the previous step, hierarchical clustering was applied to distinguish between the cytoplasm and lumen.

#### 4.2.5. Formation of Internal Gland Regions

A morphological dilation of size 3 was applied on the lumen and epithelial cells in an iterative manner until no nuclei got added to the dilated region.

#### 4.2.6. Addition of Nuclei to the Internal Glandular Region

A geometrical, heuristic approach was adopted in order to add nuclei to the identified glandular region containing lumina and epithelial cell cytoplasm. Radial lines were drawn from the center of lumen to each boundary point of the internal glandular region, and were extended towards the image boundary. The nuclei lying closest to the lumen on each radial line were considered candidate boundary objects (shown in green).

#### 4.2.7. Removal of False Nuclei

To remove false nuclei (shown in yellow), all the candidate boundary nuclei were clustered into two classes (outliers and real boundary nuclei) by applying a hierarchical clustering method on features extracted from each identified nucleus. In particular, the difference between Euclidean distance of a certain nucleus and its 5 neighboring nuclei was used in hierarchical clustering to identify real nuclei.

#### 4.2.8. Dilation of Nuclei

Once the boundary nuclei were identified, the inner gland regions were dilated until all the voxels of all the boundary nuclei were added to the inner region, thereby constituting the complete gland region.

To quantitatively measure the success of the obtained segmentation results, true+, false+, false– and true– were calculated using the manual segmentation as the gold standard and then the sensitivity, specificity, accuracy, dice similarity coefficient and Jaccard Index were computed.

### 4.3. Features for Colon Cancer Detection/grading

For each image, we extracted multiple radiomic features captured at various scales, i.e., (i) image-, (ii) gland-, and (iii) patch-based features, in order to capture various phenotypic characteristics of the tumors. 

#### 4.3.1. Image-Based Features

The overall (global) texture of the images was quantified using Haralick texture features to appreciate significantly different texture of normal and malignant colon tissues. These features were computed from the Spatial Gray Level Dependence (SGLD) matrix constructed from the input image as a whole. The SGLD matrix measures the frequency of co-occurrences of pairs of gray-levels at certain offset *d* and angle *θ*. In this work, an offset value of 1 was used at four different angles (0°, 45°, 90°, and 135°) to produce different SGLD matrices of 8 × 8 dimensions (assuming 8 scaled gray levels in the input image). Later on, the texture features of entropy, energy, correlation, inverse difference moment, inertia, sum average, sum entropy, sum variance, difference variance, difference average, and difference entropy were computed by averaging features from all the generated matrices [32].

#### 4.3.2. Gland-Based Features

The structure and shape of individual glands, which significantly vary between normal and malignant colon tissues, was also quantified to better characterize the heterogeneity by selecting the 5 largest and 5 smallest glands (in terms of area). The features extracted from each gland include: (i) Std. deviation of distances of epithelial cells (white) and cell nuclei (purple) from lumen (Figure 7b,c). The distances are computed from centroid of each object; (ii) Standard deviation of areas of epithelial cells and cell nuclei; (iii) Ratio of the sizes of different gland components (lumen, epithelial cells, nuclei) to the total pixels of the gland, and the pair-wise ratio of all the gland components (Figure 7a); and (iv) morphological features of area, compactness, convex area, eccentricity, Euler number, major- and minor-axis length, orientation and perimeter [8].

#### 4.3.3. Patch-Based Features

To extract an intermediate level of detail between the image- and gland-based features, the input image was divided into multiple patches of fixed sizes [33,34]. Random subsets of these patches were used to compute the same texture features as those of image-based features.

### 4.4. Hierarchical Classification Framework 

Support vector machines, extensively used in the past in clinical studies [35,36], were used to construct various classifiers employed in a hierarchical classification framework used in this study (Figure 2). In the first classification layer, patch-based features from subsets of patches were used to develop multiple weak classifiers based on the radial basis function (RBF) kernel of SVM. The probability estimate of the weak classifier (ProbabilityScore_2) with the highest confidence interval was used in the next step. The other individual features, i.e., image- and gland-based features, were directly classified by using the RBF kernel of SVM and the output probability estimates (ProbabilityScore_1 and ProbabilityScore_3) were used in the next step. In the second classification layer, another ensemble classification model (meta-classification) was built using linear, RBF and sigmoid kernels of SVM as weak classifiers by using probability estimates (ProbabilityScore_1, ProbabilityScore_2, and ProbabilityScore_3) as features. The output of these weak classifiers was combined through majority voting providing the final classification. For all the classification experiments within each dataset, 10-fold cross-validation was used and system parameters were adjusted based on the training data using a grid search mechanism. For quantitative performance evaluation, multiple assessment measures were computed in order to validate the reliability of results.

We have selected the top-most 5 features based on the weight assigned by the classifier, and compared the values of these features in normal and malignant groups. Boxplots of these features based on data from all the patients were created with the group type on x-axis and its value on y-axis (Figure 5). The p-values were then calculated using Wilcoxin rank test [37], and corrected using Bonferroni correction [38]. 

## 5. Conclusions

We present here an ensemble classification framework based on robust image features for accurate colon cancer detection and grading. The proposed methodology was validated both within and between two colon cancer datasets by investigating the performance of individual as well as ensemble classifiers. Our results show that the ensemble classification methodology produces robust classification and reinforces the performance of individual classifiers by a significant margin. The reliability of the proposed methodology between the two datasets suggests broader applicability across diverse clinical settings. With further prospective validation, our method may prove to be a useful tool to quickly, accurately, and reproducibly analyze colon cancer specimens.

## Figures and Tables

**Figure 1 cancers-11-01700-f001:**
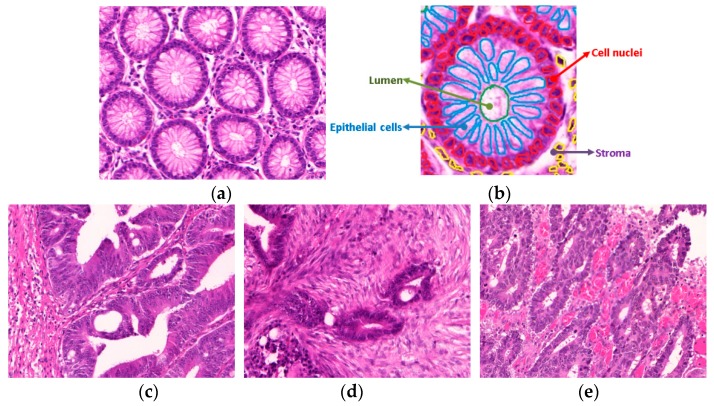
Example images: (**a**) Normal colon tissue, (**b**) detailed structure of a normal colon tissue; malignant colon tissue, (**c**) moderately differentiated, (**d**) moderately-to-poorly differentiated, and (**e**) poorly differentiated. These tissues have been stained with hematoxylin and eosin (H&E) stain; the hematoxylin stains cell nuclei purple, and eosin stains the extracellular matrix and cytoplasm pink, with other structures taking on different shades, hues, and combinations of these colors. (**a**,**c**–**e**) were captured at 20× magnification, and (**b**) is one glandular region cropped from image (**a**).

**Figure 2 cancers-11-01700-f002:**
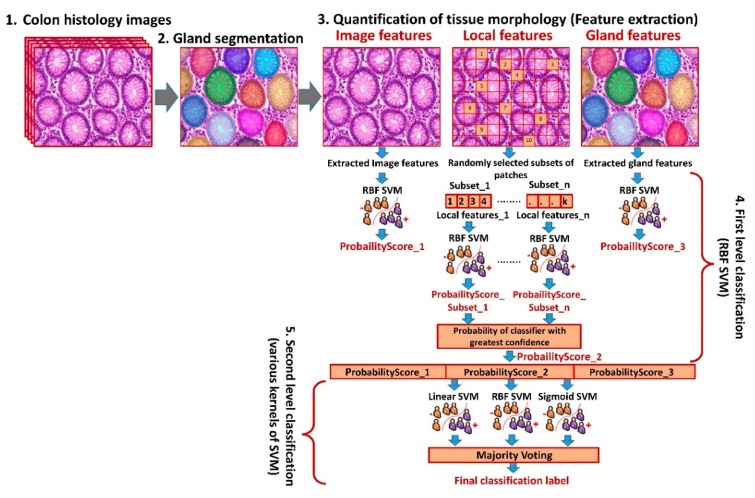
Schematic overview of the proposed methodology: (**1**) Colon tissue histology images, (**2**) gland segmentation using the proposed multi-step method, (**3**) quantification of tissue morphology, (**4**) first level of classification using radial basis function (RBF) kernel of support vector machine (SVM), and (**5**) second level of classification using majority voting based on the predictions of various weak classifiers such as linear, RBF, and sigmoid kernel of SVM.

**Figure 3 cancers-11-01700-f003:**
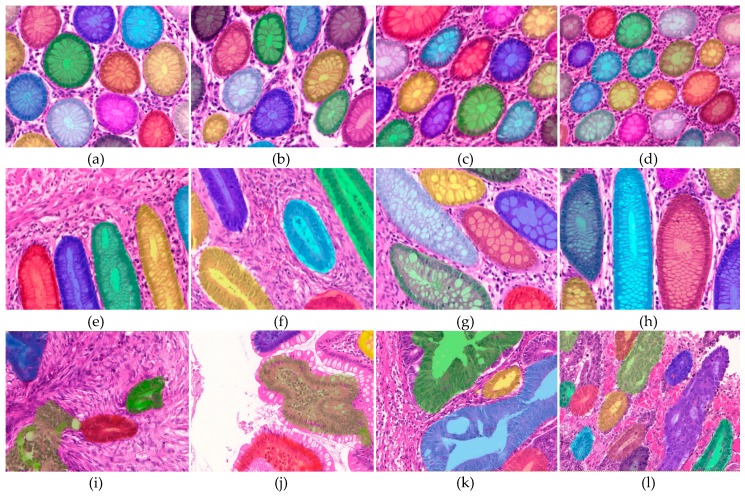
Example images showing the output of gland segmentation method. 1st row: normal samples, and 2nd and 3rd row: malignant samples of varying cancer grades. The method successfully captures the bounds of glandular regions and leads to good segmentation results for images in the 1st and 2nd rows despite the fact that images have huge variations and their glands appear in less regular structures. The third row shows example cases where the method could not identify boundaries of a few glands or has under-segmented (especially in the top-left corner of i and top-middle of j). All images were captured at 20× magnification.

**Figure 4 cancers-11-01700-f004:**
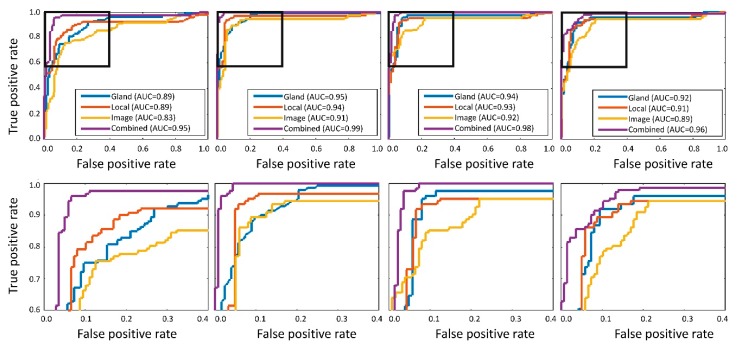
Top-row: Receiver operating characteristic (ROC) curve analysis and the corresponding area under the curves (AUCs) for individual feature categories and the meta-classifier. Bottom-row: zoomed in version of ROC curves. Left-to-right: Cancer detection for RMC-Dataset, cancer detection for GlaS-Dataset, cancer grading for RMC-Dataset, and cancer grading for GlaS-Dataset.

**Figure 5 cancers-11-01700-f005:**
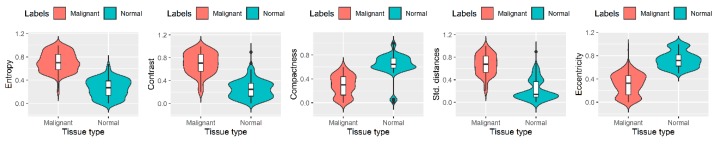
Interpretation of features of normal and malignant colon tissues. X-axis of the box-plots shows the tissue type, which is either normal or malignant, and the y-axis shows the value of the feature. The feature of standard deviation of distances (Std. distances) was scaled in the range (0–1) for better visualization.

**Figure 6 cancers-11-01700-f006:**
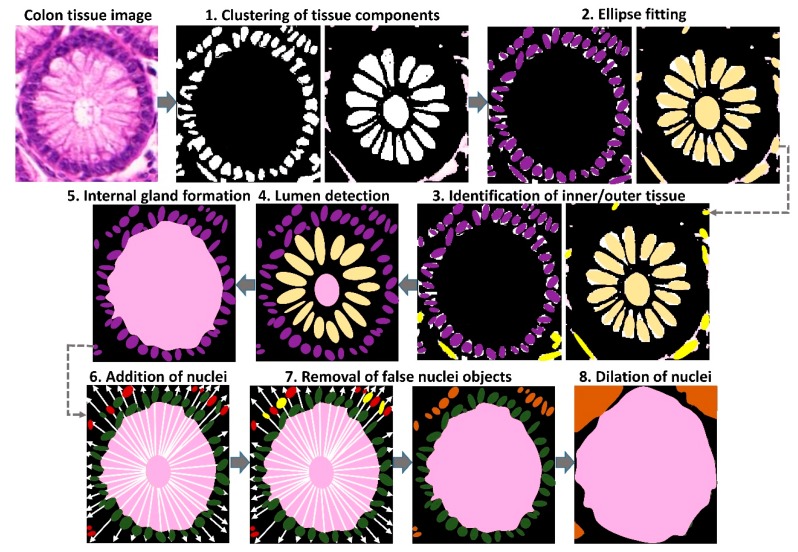
Schematic workflow of the proposed gland segmentation method. Figure shows one glandular region cropped from an image captured at 20× magnification.

**Figure 7 cancers-11-01700-f007:**
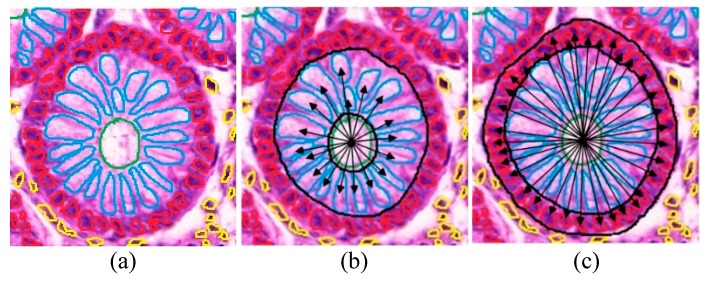
(**a**) A single normal colon gland with demarcated lumen, epithelial cells and nuclei; area features are computed from epithelial cells, lumen and nuclei, (**b**) distance from the centroid of lumen to the centroid of each epithelial cell, (**c**) distance from the centroid of lumen to the centroid of each cell nucleus. Figure shows one glandular region cropped from an image captured at 20× magnification.

**Table 1 cancers-11-01700-t001:** Performance of the proposed gland segmentation method.

Performance Metrics	RMC-Dataset	GlaS-Dataset
Segmentation accuracy	87.50	88.40
Jaccard index	0.86	0.89
Dice similarity	0.84	0.87
Sensitivity	0.90	0.92
Specificity	0.82	0.88
F-Score	0.88	0.89

**Table 2 cancers-11-01700-t002:** Performance of individual feature extraction methods (results are shown for the ensemble classifier for the patch-based features).

	Gland-Based		Local-Patch-Based		Image-Based	
	GlaS-Dataset	RMC-Dataset	GlaS-Dataset	RMC-Dataset	GlaS-Dataset	RMC-Dataset
**Cancer Detection**						
Accuracy	93.7	92.1	93.1	91.5	92.5	90.9
Sensitivity	92.4	86.8	91.3	85.2	92.3	84.5
Specificity	95.1	98.6	93.9	97.4	92.7	95.9
MCC	87.4	85.0	86.2	83.7	85.0	82.3
**Cancer Grading**						
Accuracy	90.5	92.5	89.5	91.5	89.7	90.7
Sensitivity	84.4	89.0	82.9	87.0	81.5	85.5
Specificity	93.1	94.9	92.0	92.5	91.6	92.4
MCC	79.0	83.9	76.0	80.6	74.7	78.6

**Table 3 cancers-11-01700-t003:** Performance of the meta-classifier for both data sets (results are shown for individual and ensemble classifier).

	GlaS-Dataset				RMC-Dataset			
	Linear	RBF	Sigmoid	Ensemble	Linear	RBF	Sigmoid	Ensemble
**Cancer Detection**								
Accuracy	94.3	95.4	94.8	98.3	92.1	93.3	92.7	97.6
Sensitivity	91.3	92.4	92.4	97.8	96.7	98.9	98.9	98.9
Specificity	97.6	98.8	97.6	98.8	86.5	86.5	85.1	95.9
MCC	88.7	91.2	89.8	96.5	84.2	86.9	85.8	95.1
**Cancer Grading**								
Accuracy	94.2	94.5	93.7	98.6	91.3	93.6	93.5	98.6
Sensitivity	90.7	91.4	90.4	97.3	84.7	87.7	87.8	97.4
Specificity	95.7	95.9	95.0	99.0	93.7	95.7	95.4	99.0
MCC	86.4	87.3	85.4	96.4	78.4	83.3	83.2	96.4

**Table 4 cancers-11-01700-t004:** Performance of the meta-classifier across multi-institutions (results are shown for individual and ensemble classifiers).

	Train RMC-Dataset + Test GlaS-Dataset				Train GlaS-Dataset + Test RMC-Dataset			
	Linear	RBF	Sigmoid	Ensemble	Linear	RBF	Sigmoid	Ensemble
**Cancer Detection**								
Accuracy	89.7	91.5	90.9	94.5	89.7	90.2	87.9	93.7
Sensitivity	97.8	96.7	96.7	96.7	96.0	96.7	91.3	96.7
Specificity	79.7	85.1	83.8	91.9	81.7	82.9	84.1	90.2
MCC	79.9	83.1	81.9	89.0	79.9	80.9	75.8	87.4
**Cancer Grading**								
Accuracy	89.8	91.3	90.0	95.0	89.8	90.6	90.7	95.0
Sensitivity	82.2	84.5	81.6	91.6	84.1	84.7	86.0	92.8
Specificity	92.2	93.6	93.0	96.3	92.1	92.8	92.4	95.8
MCC	74.3	78.2	74.6	87.9	76.2	77.5	78.5	88.6

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
