# Peer review of "Segmentation and Grade Prediction of Colon Cancer Digital Pathology Images Across Multiple Institutions"

_cancers, 2019, doi:10.3390/cancers11111700_

Round 1

Reviewer 1 Report

The manuscript by Saima Rathore et al., demonstrates the use of computational method using image patches in colon cancer. The group addresses a very important question for cancer detection and predicts the tumor grade on easily obtaining images of histology images.

This paper is actually a progress report on a series of papers already published by the corresponding author and should be published after my minor concern will be addressed.

As a scientific paper the work should be able to stand alone to the scrutiny of the scientific reader. As such can the authors improve and offer more details on the specific computational methods used and a clarification on how the optimal combination of futures on the meta-classifier was achieved in Fig. 5

Once the minor revision is done, the paper is suitable for publication in Cancers.

Reviewer 2 Report

The authors present an end-to-end framework for segmentation, detection and grading of cancer. This end-to-end pipeline first employs a multi-scale ellipse fitting approach. Then, gland-level features, patch-level and image-level features are utilised within support vector machine to detect cancer and diagnose the grade.

The first two things that come to mind when developing a segmentation approach are the performance and the processing time of an image. Given that the accuracy/performance is such an important constituent of any segmentation model, it is very important to first report the performance of the segmentation approach and then give comparison to other methods. In particular, why was this method used as opposed to a deep learning alternative? Was it beneficial in terms of the processing time? Full justification is needed.

Following on from the above, the literature review particularly for gland segmentation is insufficient. There are plenty of recent top performing approaches following on from the Kainz et al. paper that the authors chose to compare with. In particular, it is very important to mention DCAN (the winner of the GlaS challenge) and some other approaches. I would recommend to add the following:

Chen, Hao, et al. "DCAN: Deep contour-aware networks for object instance segmentation from histology images." Medical image analysis 36 (2017): 135-146. Ronneberger, Olaf, Philipp Fischer, and Thomas Brox. "U-net: Convolutional networks for biomedical image segmentation." International Conference on Medical image computing and computer-assisted intervention. Springer, Cham, 2015. Graham, Simon, et al. "MILD-Net: Minimal information loss dilated network for gland instance segmentation in colon histology images." Medical image analysis 52 (2019): 199-211.

I assume that this gland segmentation approach is used because not only the gland is segmented, but the nuclei and lumen are additionally segmented that are then used within the SVM for grading. If that is the case, it needs to be made clear. I suggest that this also is added to the motivation for using this gland segmentation approach.

I also think that the cancer grading literature review is insufficient. For example, the below paper tackle the problem of grading cancer:

Awan, Ruqayya, et al. "Glandular morphometrics for objective grading of colorectal adenocarcinoma histology images." Scientific reports 7.1 (2017): 16852.

I’m assuming that cancer detection is classifying between normal and cancer; then cancer grading is then breaking down the malignant cases into their grades. This needs to be explained more clearly. Please put this up front in the paper.

In the tables, only the accuracy has been put in bold; I am assuming that the maximum value in each row should be put in bold.

I think that the idea of combining the different features from the individual glands and the images is the most significant contribution of this paper. However, I don’t think multi-scale is the best way to describe this approach. Instead I would suggest using: gland, local and image level features. To highlight this I would improve Figure 2 - make it more compact. Also, I am assuming that the red/green/blue rectangles highlight the extracted features but this is not clear at all! Instead, include a phrase highlighting the feature extraction technique.

Minor remarks:

Use GlaS (Gland Segmentation) not GLas. This is minor, but should be fixed when used throughout the paper.

Please refer to Heamatoxylin and Eosin rather than ‘purple and ‘pink’. A line or two can be added to the introduction and also to the dataset section.

The legend is cut off in Figure 4. I also recommend using a zoom box of the top left corner of each ROC curve.

Overall, the paper has potential but certain issues need to be addressed before publication. Main points:

Improve the literature review Give justification for the proposed gland segmentation approach Focus on the ordering of the article - make it clear what is the task from the beginning. I.e what is the difference between detection and grading.
